mechanical engineering

damage assessment, orthogonal decomposition, real-time monitoring, composite materials, digital image correlation

**Author for correspondence:**
W. J. R. Christian
e-mail: w.j.r.christian@liverpool.ac.uk

# Real-time quantification of damage in structural materials during mechanical testing

W. J. R. Christian[1], K. Dvurecenska[1], K. Amjad[1], J. Pierce[2], C. Przybyla[3] and E. A. Patterson[1]

[1]School of Engineering, University of Liverpool, Liverpool, UK
[2]University of Dayton Research Institute, Dayton, OH, USA
[3]Air Force Research Laboratory, Wright-Patterson AFB, OH, USA

WJRC, 0000-0003-3638-7297; KA, 0000-0002-9348-0335;
EAP, 0000-0003-4397-2160

A novel methodology is introduced for quantifying the severity of damage created during testing in composite components. The method uses digital image correlation combined with image processing techniques to monitor the rate at which the strain field changes during mechanical tests. The methodology is demonstrated using two distinct experimental datasets, a ceramic matrix composite specimen loaded in tension at high temperature and nine polymer matrix composite specimens containing fibre-waviness defects loaded in bending. The changes in the strain field owing to damage creation are shown to be a more effective indicator that the specimen has reached its proportional limit than using load-extension diagrams. The technique also introduces a new approach to using experimental data for creating maps indicating the spatio-temporal distribution of damage in a component. These maps indicate where damage occurs in a component, and provide information about its morphology and its time of occurrence. This presentation format is both easier and faster to interpret than the raw data which, for some tests, can consist of tens of thousands of images. This methodology has the potential to reduce the time taken to interpret large material test datasets while increasing the amount of knowledge that can be extracted from each test.

## 1. Introduction

Composite materials have complicated heterogeneous microstructures that typically contain many microscale defects, this results in complicated damage mechanics that are difficult

to predict. While some knowledge can be gained from studying components after failure, the inherent complexity of composites makes it difficult to determine the mechanisms involved in transition from the manufactured state to a failed state. To study this process, typically specimens are monitored while they are loaded to failure. The challenge then becomes how to maximize the information recorded for each specimen while ensuring that the data can still be interpreted. Several techniques for capturing data during mechanical testing are available within two broad categories: contact methods, such as strain gauges or monitoring failure based on the propagation of sound waves; and non-contact methods, which typically use visible light-based measurements or computed tomography. Strain gauges provide pointwise measurements and thus to monitor an area and locate damage would require many gauges, which would lead to excessive wiring and instrumentation. Therefore, acoustic emission is a commonly used contact method for detecting the onset of damage [1] and can also be used to triangulate where damage is occurring in specimens [2]. Another approach is monitoring the propagation of Lamb waves from transducers to sensors [3]. The propagation of sound is affected by the presence of damage and thus the damage can be detected and approximately located. A drawback of these techniques is that they require sensors to be in contact with the specimen. This imposes limitations on the size of specimens as well as the conditions in which they are tested. Non-contact methods such as computed tomography are capable of determining the morphology of damage as it forms in specimens [4,5] but these techniques require expensive infrastructure restricting the number of specimens and the rate of testing.

Methods based on visible light use equipment that is less expensive and thus more readily available. For translucent specimens, such as glass-fibre reinforced polymers, subsurface cracks can be automatically identified during mechanical tests by monitoring white light passing through the specimen [6,7]. This allows images of damage morphology to be created during tests but cannot be applied to opaque materials. Full-field optical measurements, such as digital image correlation (DIC), can be used to monitor the strain field on the surface of any structural material, but without the requirement for excessive wiring and instrumentation that is inherent with the use of strain gauges. The strain fields obtained are often used to monitor for strain hot-spots that indicate locations where damage is being created [8,9]. However, this does not make full use of the quantitative data that can be obtained from full-field techniques. Statistical approaches, which indicate the severity of damage, have been applied to data from Doppler vibrometry [10] and thermoelastic stress analysis [11], but these techniques require cyclical loads in order to perform measurements.

A further issue with all monitoring techniques is that as the complexity of the damage progression increases the number of images or samples required to resolve it increases. For example, to monitor rapid damage events such as impact, high-speed imaging must be used, resulting in thousands of strain fields showing specimen behaviour [12]. This greatly increases the amount of time required to analyse experimental data. One approach to this problem is to reduce the amount of time spent interpreting each image. Clustering is an approach that has been used to automate the identification of slip systems in large strain fields on the surface of metals [13]; while the virtual fields method can be used to identify reductions in stiffness in specimens based on strain fields, resulting in images that indicate the presence and approximate location of damage [14]. These techniques make it easier for engineers to interpret large datasets but do not eliminate the need to analyse each captured strain field. Another approach is to monitor the strain or displacement as a function of time at particular locations on the surface of a specimen using DIC [12], but this discards much of the full-field data. Orthogonal decomposition is a technique that has been used to reduce full-field data down to a small number of coefficients that succinctly describe surface deformation [15]. This has been used to compare measurements from dynamic experiments with predictions from finite-element models such that the differences between the two datasets can be displayed as line traces [16]; and recently, has been used to compare damaged specimens with virgin specimens for quantitative damage assessments [17]. The decomposition-based damage assessment compared the strain field on the surface of a damaged specimen with the strain field of an undamaged specimen, resulting in a defect severity metric that was found to correlate with the residual strength of the damaged specimen. However, this technique cannot be used for monitoring the progressive creation of damage in a specimen as it requires a reference strain field captured at the same load as the damaged strain field in order to make comparisons. In this study, the rate of change of the strain field is monitored instead. This removes the need for a reference strain field while still allowing damage to be quantified. The technique is applied to two exemplars: a ceramic matrix composite (CMC) specimen loaded in tension at high temperature and nine polymer matrix composite (PMC) specimens loaded in bending.

## 2. Strain-based damage monitoring

In a material test, as the load is increased, damage is initiated in the material; when this damage forms, it acts as a stress concentration resulting in a measurable change to the strain field. If a DIC system is used to measure the surface first principal strain field during the material test, then changes owing to damage initiation and the time at which these changes occurred can be determined. Initially during a test, the measured strain field only changes because of elastic deformation and measurement noise; however, when damage is created, the strain field changes significantly and permanently. It is these significant changes that can be used to detect when damage occurs, where it occurs and its severity. The measured strain fields typically contain large amounts of redundant data. This means that each strain field is described by a greater amount of data than was needed for analysis. This excess data increases the amount of memory required to store the measurements, as well as the computation time required when processing it. In this work, orthogonal decomposition [15] was used to reduce these data, such that each strain field was represented by a relatively small number of coefficients collated into a feature vector. This was achieved by projecting each strain field onto a set of discrete Chebyshev polynomials [18]. The feature vectors could be reconstructed back into a strain field to ensure that they are a reasonable representation of the original data. In theory a perfect reconstruction is obtained if the number of coefficients is equal to the number of pixels in the original strain field, however as the strain fields are typically smooth with few discontinuities an accurate reconstruction can be obtained with just a small number of coefficients. Hence, when the representation error is equal to the measurement uncertainty of the DIC system, the key information about the strain field is captured while the redundant information is rejected. The number of coefficients used for each dataset is dependent on the complexity of the patterns in the strain fields.

Once decomposed, the dissimilarity between two strain fields was quantified by considering the feature vectors for each strain field as defining locations in multi-dimensional space. The distance between these locations were then calculated using the Euclidean distance. This provides a scalar value that represents the dissimilarity between the strain fields. The rate of change of the strain field was calculated by dividing the Euclidean distance by the time difference between the instant of capture of the two strain fields:

$$\text{rate of change} = \dot{s}(t) = \frac{\|f(t) - f(t-h)\|}{h}, \tag{2.1}$$

where $f(t)$, is the feature vector for the strain field captured at time $t$ and $\| \cdot \|$, is the vector norm. The time difference $h$, controls the time range over which the rate of change is calculated, with higher values resulting in less noise. This parameter should be greater than the time period between camera frames. Equation (2.1) is evaluated as each strain field is captured, this means that if an instantaneous event occurs that causes the strain field to permanently change then the rate of change is increased for multiple samples. This behaviour was used to identify when damage events occur and is described later in this section.

Changes in the measured strain field also occur owing to elastic deformation of the specimen and measurement noise. These changes can be considered as a baseline because they are reasonably constant during the test, if the rate of change is significantly above this baseline then the strain field is being permanently changed owing to damage. The baseline can be estimated using one of two different methods. Firstly, if displacement control is used in the test and the relationship between applied displacement and gross strain is linear, then the baseline will be constant. This is because the rate of elastic deformation will be constant. This behaviour is typical of most structural materials loaded in tension. In this situation the baseline can be determined by calculating the mean of the rate of change, $\dot{s}(t)$ during the initial stages of the experiment, before damage is created in the specimen. When deformations are large the relationship between applied displacement and gross strain is often nonlinear [19] and thus the baseline value of the rate of change will change with time. In this situation a second method of estimating the baseline can be used. This method exploits the behaviour of composites, in which damage events are typically discrete and thus the rate of change, $\dot{s}(t)$ increases only momentarily during the event before returning back to the baseline. This means the baseline can be estimated by calculating the moving median of the rate of change, $\dot{s}(t)$. The moving median is used instead of the moving mean as it is robust to outliers in the rate of change data [20], such as when damage events occur. The period of time over which the median is calculated must be greater than the time difference $h$, used to calculate the rate of change. This is because the width of the peaks in the rate of change data owing to damage events will be equal to $h$. A value of $3h$, has

been found to result in a good estimate of the baseline value of $\dot{s}(t)$, this is based on observations of the fit between the baseline and the rate of change data. Once the baseline value is estimated, it can be subtracted from the rate of change data resulting in an indication of the rate of damage creation, in terms of how the strain field changes:

$$\text{indicated damage rate} = \dot{s}_d = \dot{s} - \dot{s}_b, \qquad (2.2)$$

where $\dot{s}_b$ is the baseline rate of change. The amount by which damage had changed the strain field, hereafter termed *indicated damage severity*, was calculated by integrating equation (2.2) with respect to time using the trapezium rule:

$$\text{indicated damage severity} = s_d = \int_0^t \dot{s}_d \cdot \mathrm{d}t. \qquad (2.3)$$

This quantity has the units of strain and provides an estimate of the accumulated change in the strain field since the start of the test owing to the initiation and propagation of damage. These fundamental steps in evaluating the severity of damage are shown in the flowchart in figure 1. Damage events, e.g. matrix cracking or delamination propagation, were identified by looking for peaks in the indicated damage rate from equation (2.2). However, it can be difficult to automatically identify significant events when the data contains substantial amounts of measurement noise. This is because noise causes the rate of change, $\dot{s}(t)$ to occasionally peak, but these peaks only occur for a single measurement; whereas, when damage forms, the strain field at that location is permanently changed. As the rate of change was calculated over a time difference $h$, significant damage events cause the indicated damage rate to peak for a time period equal to $h$. This is because $h$ is an integer multiple of the time period between the acquisition of DIC images and thus an instantaneous change to the strain field affects multiple rate of change samples. Events that caused significant changes to the strain field were detected by looking for these sustained peaks. A threshold for identifying peaks was set to the 95th percentile of the indicated damage rate during a time period in which damage events were not expected to occur. This can either be determined from similar tests where damage events do not occur or can be measured in the early stages of a test, before damage is initiated.

Maps showing the location and time of damage events, referred to as damage-time maps, were created using pairs of feature vectors from just before and just after damage events were detected. These maps were created using an iterative algorithm. At the start of a test the specimen can normally be assumed undamaged, therefore, a blank damage-time map was initially defined in the computer memory. This initial damage-time map was then updated using each strain field in the time sequence. The updating algorithm is shown as a flow diagram in figure 1 and described in detail here. The strain field that was being processed during each iteration was first decomposed into a feature vector, $f(t_i)$, where $t_i$, is the time at which the strain field was captured. The difference between this feature vector and a feature vector at an earlier time, $f(t_i - h)$, was then calculated. This resulted in a strain-difference feature vector that represented the recent changes in the strain field. The indicated damage rate was calculated from the strain-difference feature vector using equations (2.1) and (2.2) and checked to see if a damage event was occurring. If a damage event was occurring then the strain-difference feature vector was reconstructed to yield a strain-difference field, showing the locations where strain had changed and the magnitude of the changes. In this field, damaged regions had higher strains than undamaged regions because the damage acts as a stress concentration. These high strain regions were identified by thresholding the strain-difference field. The threshold was determined by calculating the 99.9th percentile of the strain-difference field represented by the feature vector, $f(h) - f(0)$. The changes in this field occurred at the start of the test and thus were only caused by measurement noise and elastic deformation. After thresholding, the strain-difference field indicated regions that were high strain and regions that were low strain. The pixels in the damage-time map that corresponded to the high strain regions in the strain-difference field were given a value corresponding to the capture time for the currently processed strain field, $t_i$. If any of these pixels had already been assigned a value in a previous iteration then the time value in those pixels was unchanged. At the end of the iteration the damage-time map can be displayed to show the locations where damage was likely to have occurred up until the time, $t_i$. The algorithm was then repeated with each strain field until all strain fields had been processed. After all of the strain fields were processed, the finished damage-time map showed the locations and times at which damage first occurred in the specimen.

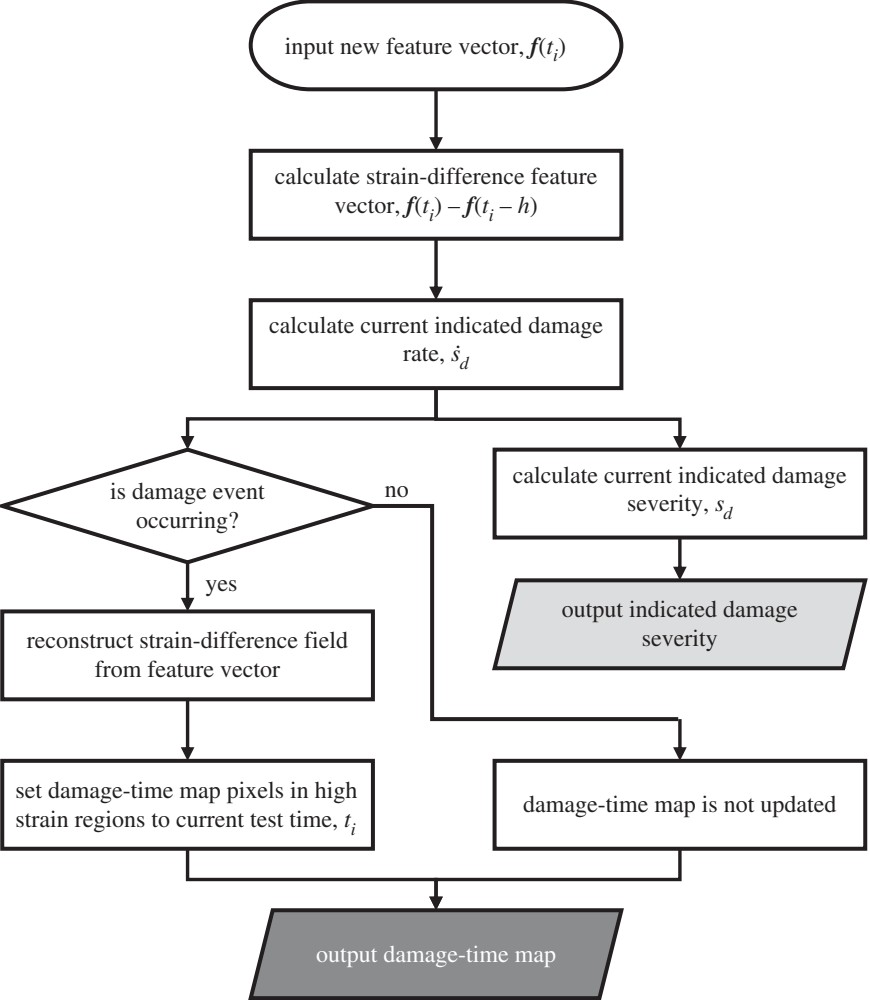

**Figure 1.** Flow diagram of the algorithm for updating the damage-time maps (dark grey output) and the indicated damage severity (light grey output). The algorithm was applied to feature vectors representing the strain fields.

The algorithm described in this section was applied to data captured from two distinct tests, a $SiC_f/$ SiC dogbone specimen undergoing tensile loading, referred to as the CMC specimen, and multiple four-point bend experiments on specimens of PMC with defects, referred to as the PMC specimens. These experiments are described in the next section.

# 3. Experimental method

## 3.1. Ceramic matrix composite experiment

A single CMC specimen was quasi-statically loaded to failure in tension at high temperature. The specimen was made of HiPerComp™ (GE Aviation, USA), consisting of unidirectional plies with a $[90/0]_{2s}$ layup, with the 0° plies orientated along the longitudinal direction of the specimen. The gauge section was 60 mm long with a cross-section measuring 8 mm by 2.2 mm. A speckle pattern was applied by first spraying the specimen black (HIE-Coat 840-C, Aremco, USA) before using an air gun to apply a diluted alumina paste (Ceramabond 569, Aremco, USA) to form the pattern. Prior to loading, the specimen was heated to a temperature of 1100°C by a 1 kW continuous-wave laser, without using a chamber. Once at the required temperature, the specimen was loaded at a rate of 1.5 mm min⁻¹. A Vic3D stereographic DIC system (Correlated Solutions, USA) was used to capture images of the specimen during loading. The experiment was conducted by the Air Force Research Laboratory (AFRL) at Wright-Patterson Air Force Base in Ohio, but the images were processed using Istra4D (Dantec Dynamics, Germany) at the University of Liverpool. The parameters used to perform

**Table 1.** Parameters used to perform DIC analysis on the CMC specimen.

| DIC hardware parameters | |
| --- | --- |
| camera make | Grasshopper3 |
| camera manufacturer | FLIR, USA |
| sensor size | 2448 by 2048 pixels |
| lens make | Micro-Nikkor, 25 mm |
| lens manufacturer | Nikon, Japan |
| field of view | 96.2 mm |
| image scale | 32.3 pixels mm$^{-1}$ |
| stereo angle | 25° |
| stand-off distance | 167 mm |
| acquisition rate | 1.33 Hz |
| nominal speckle size | 0.10 mm |
| DIC software parameters | |
| subset size | 15 pixels |
| step size | 10 pixels |
| subset shape functions | second order polynomial |
| measured quantity | first principal strain (Lagrangian) |
| strain window | 3 by 3 subsets |
| virtual strain gauge size | 35 by 35 pixels |

the DIC analysis are listed in table 1. These parameters were chosen to maximize the spatial resolution of the first principal strain data such that crack formation could be observed and measured.

Because the gauge region of the specimen had a high aspect ratio, the strain field was split into six separate regions measuring 7.0 mm by 9.5 mm, which are shown in figure 2. Each region was orthogonally decomposed into feature vectors containing 235 coefficients, resulting in a relative representation error of 7%. This number of coefficients was arbitraily chosen as the measurement uncertainty of the experiment conducted at AFRL was unknown. The relative representation error was obtained by first calculating the root mean square of the difference between the original and reconstructed strain field, this quantity is the representation error, and then dividing it by the range of the original strain field. Clusters of adjacent pixels where the difference between the reconstruction and the original strain field was greater than three times the representation error were identified to test the local reconstruction quality. All clusters were smaller than 0.3% of the total number of pixels, indicating that there were no significant localized areas where the reconstruction was poor. These feature vectors were then used as the input for the algorithm described in the previous section. The value of each rate of change, $\dot{s}(t)$ which was obtained from equation (2.1), was calculated using a time difference, $h$, of 3 s. This is equivalent to the time taken for five frames to be captured. The baseline rate of change was assumed to be constant as the specimen was loaded in tension and the deformation was small. This constant baseline was estimated by calculating the mean of the rate of change during the first 100 s of the test, before the specimen reached its limit of proportionality.

## 3.2. Polymer matrix composite experiment

The four-point bend experiments were conducted at the University of Liverpool by a different experimentalist to the one that conducted the experiment described in §3.1. Nine specimens were produced with the dimensions 40 mm by 220 mm, and a thickness of 2.9 mm. The specimens were made of unidirectional prepreg plies (RP507, PRF, UK) with a $[0_2/90_2/-45_2/45_2]_S$ layup, where 0° indicates that the fibres run in the longitudinal direction of the specimen. The plies were laid up over special formers that have the appearance of a small gable roof with a rounded ridge. As the plies were laid over the ridge of the former, the fibres that were on the top surface of the specimen were slightly longer than the fibres on the bottom surface. When the specimen was removed from the

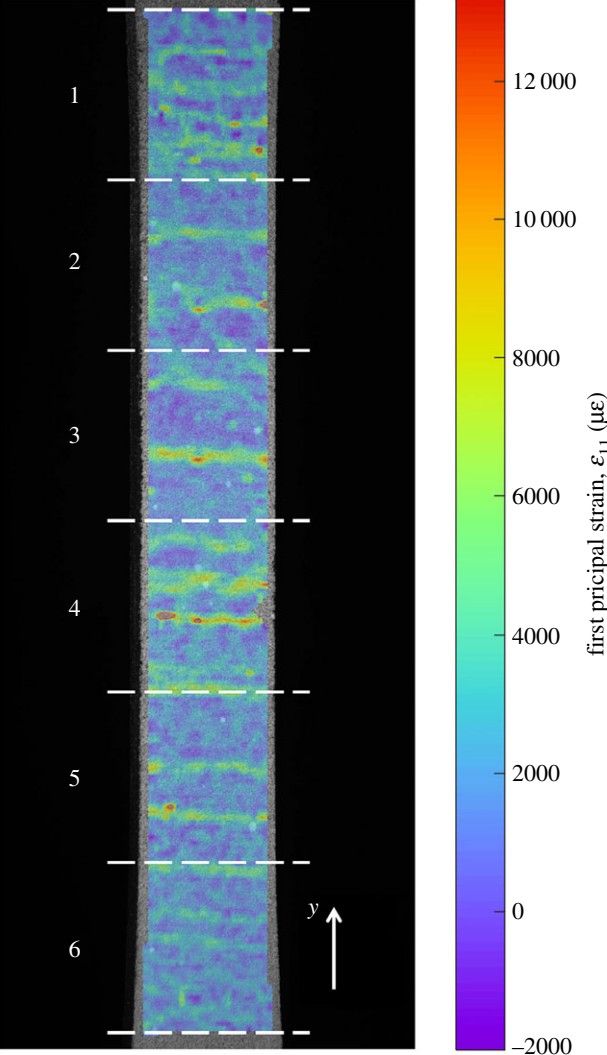

**Figure 2.** The first principal strain field on the surface of the CMC specimen at a load of 4 kN. The six regions are indicated by the white dashed lines and the y-direction for the data is indicated in the bottom-right corner. This figure is best interpreted in colour.

former and then flattened, the longer fibres on the top surface were put into compression and thus buckled, this resulted in a waviness defect at the centre of the uncured specimen on its top surface. The percentage change in length of the fibres from their unbuckled to their buckled state was used to characterize the defect severity, this is referred to as nominal waviness. The nominal waviness is dependent on the dimensions of the former used during layup. The equations used to design the formers are described in previous work by the authors [21], which describes how waviness could be created in specimens, but only qualitatively explored the damage mechanics of these specimens. For this experiment three specimens were produced at each of the following nominal levels of waviness: 0% (i.e. defect free), 20% and 25%. The laminates were then cured in a hot press (APV-2525, Meyer, Germany) at a temperature of 130°C and pressure of 2.1 bar for 45 min. Finally, the specimens were cut to the dimensions shown in figure 3 using a wet diamond saw (Versatile Power Pro 900, Vitrex, USA).

The specimens were loaded in a bending rig with a support span of 160 mm and a loading span of 80 mm. The specimens were loaded to failure at a crosshead rate of 0.8 mm min$^{-1}$ with the defective ply in compression and with the defect at the centre of the load span. A speckle pattern was applied using aerosol spray paints on the tensile side of each specimen enabling DIC to be used to measure surface strain during loading. A stereoscopic DIC system (Q-400, Dantec Dynamics, Germany) was used to measure the first principal strain on the surface of each specimen. The parameters used for the DIC analysis are listed in table 2. A calibration experiment was conducted to estimate the measurement uncertainty of the DIC system using an aluminium specimen with a bonded strain gauge and similar quality of speckle pattern to the ones on the PMC specimens. Firstly, the specimen was placed in the

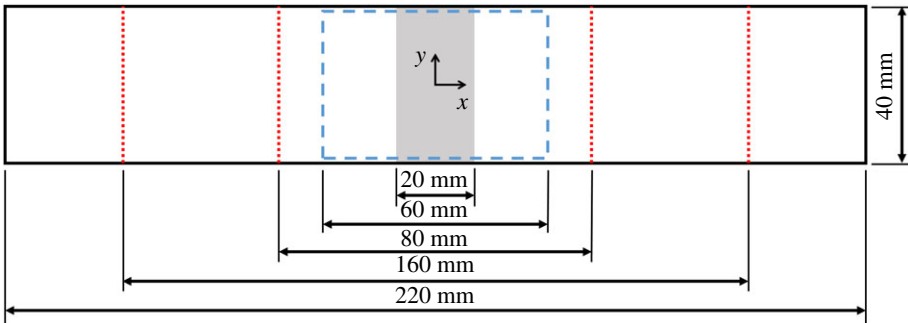

**Figure 3.** Diagram of the PMC specimens showing the region containing waviness (shaded grey), the region of interest for the DIC system (dashed box), and the locations at which out-of-plane loads were applied using the four-point bending rig (dotted lines).

**Table 2.** Parameters used to perform DIC analysis on the PMC specimen.

| DIC hardware parameters | |
| --- | --- |
| camera make | Stingray F125 |
| camera manufacturer | Allied Vision, Germany |
| sensor size | 1292 by 964 pixels |
| lens make | Cinegon 1.4/12-0906, 12 mm |
| lens manufacturer | Schneider, Germany |
| field of view | 135 mm |
| image scale | 15.5 pixels mm$^{-1}$ |
| stereo angle | 63° |
| stand-off distance | 204 mm |
| acquisition rate | $66.6 \times 10^{-3}$ Hz |
| nominal speckle size | 0.25 mm |
| DIC software parameters | |
| subset size | 25 pixels |
| step size | 10 pixels |
| subset shape functions | second order polynomial |
| measured quantity | first principal strain (Lagrangian) |
| strain window | 11 by 11 subsets |
| virtual strain gauge size | 125 by 125 pixels |

bending rig and a bending moment applied to it, DIC was then used to measure the first principal strain on the surface of the specimen. The experiment was performed six times and the root mean squared difference between the DIC measurements and the strain measured by the strain gauge was recorded. This resulted in an estimate of measurement uncertainty of 44 µε.

Each strain field was orthogonally decomposed into feature vectors containing 235 coefficients using Chebyshev polynomials. Each vector was then processed to set coefficients with low absolute values to zero. The threshold indicating that the coefficients were low was chosen using an iterative algorithm that calculated the highest threshold at which the representation error after reconstruction was less than the measurement uncertainty of the DIC system [17]. After processing, the feature vectors contained fewer than 40 non-zero coefficients. The local reconstruction quality was also checked and found to be acceptable using the same criterion described in §3.1. Each value of the rate of change, $\dot{s}(t)$ was calculated using a time difference of 30 s, equivalent to the time taken for three frames to be captured. The baseline for the rate of change data varied owing to the large out-of-plane deformations of the specimens. Therefore, the baseline value for the rate of change was estimated using the moving-median method described in the previous section.

# 4. Results

The CMC specimen was tested in tension with a constant rate of crosshead displacement. Therefore, while deformation was elastic the load increased linearly with time but as the specimen started to permanently deform, the relationship between measured load and time became nonlinear. The time at which this transition occurs, equivalent to the yield point in metals but referred to here as the limit of proportionality, can be identified using the plot of the indicated damage severity, $s_d$ obtained using equation (2.3). The value of the indicated damage severity is nominally constant for the first half of the test before rapidly increasing after $t = 170$ s. This can be seen in figure 4. The local variations in the indicated damage severity can be used to identify which regions of the specimen contain the most severe damage. For example, figure 4 suggests that damage is created in region no. 4 at almost twice the rate in region no. 5.

Damage events can be detected by monitoring for peaks in the indicated damage rate, $\dot{s}_d(t)$ obtained using equation (2.2). These events can be used to construct damage-time maps showing the position and time at which damage occurs. The damage-time map for region no. 4 of the CMC specimen is shown in figure 5. The damage-time maps for all of the regions can be stitched together to yield a single damage-time map for the whole specimen, this is shown in figure 6.

The PMC specimens were loaded in bending with a constant displacement rate applied by the bending rig supports. The applied bending moment was calculated from the dimensions of the bending rig and the compressive load measured by the test machine on the loading noses of the bending rig. While the deformations were low, the relationship between measured moment and crosshead displacement was linear, and thus the relationship between measured moment and time was also linear. As the deformation of the specimen became larger the relationship ceased to be linear, to show this behaviour a line of best fit was added to the moment-time curve for a 0% nominal waviness specimen in figure 7. To account for this non-linearity the moving median method was used to estimate the baseline rate of change of the strain field. Figure 8 shows how different choices of the time period over which the median is calculated affects the indicated damage severity data. If the time period is too long, or a constant baseline is assumed, then in the later stages of the test, the baseline rate of change is over-estimated and the indicated damage severity reduces, suggesting that the specimen is becoming less damaged. This is not possible and thus illustrates the importance of techniques for estimating the baseline when it is likely to vary. Other methods of removing the baseline may be possible, the moving median and constant baseline approaches were chosen for this study as they relied on simple statistics.

This indicated damage severity data can be used to identify when delaminations start to form in the PMC specimens. At the top of figure 9, the measured moment initially increases linearly with time until about 720 s. Soon after this instant, a significant reduction in stiffness occurs as delaminations form and grow. This event can be observed in the indicated damage severity data, which at the same time significantly increased. A second reduction in stiffness occurs at 1050 s, when the moment carried by the specimen dropped to below 12.75 Nm (equal to 50% of the peak moment), triggering the loading machine to stop. This reduction in stiffness was also observed in the indicated damage severity data. Spatio-temporal information about the damage propagation in the PMC specimens was obtained by calculating their associated damage-time maps. In the middle of figure 9, a single DIC strain field captured at 700 s is shown. At the centre of this strain field is a strip of high strain caused by the misaligned fibres or waviness at this location. In the damage-time map at the bottom of figure 9, this location is shown to be the initiation point for damage which progressively grows in the negative $x$-direction from the specimen centre before a sudden increase in the positive $x$-direction at 1050 s.

All, with one exception, of the PMC specimens containing nominal values of waviness of 20% and 25%, underwent the same path to failure, in which damage first progressed on one side of the defect before progressing on the opposite side. The exception was a single specimen with a nominal waviness of 20%, in which a delamination suddenly formed causing the stiffness of the specimen to drop by 50% and thus triggered the test to end. The damage-time map for this specimen indicated that the damage only covered approximately one-third of the region of interest. The size and shape of this delamination was determined by pulse-echo ultrasonic non-destructive evaluation using the procedure described in [22] to yield a plan view of the damage. This allowed a comparison of the damage morphology indicated in the damage-time map with traditional non-destructive measurements of the damage. The resulting time-of-flight ultrasound data is shown in figure 10. The image shows the depth in the material of the first surface to reflect ultrasound, which is the bottom

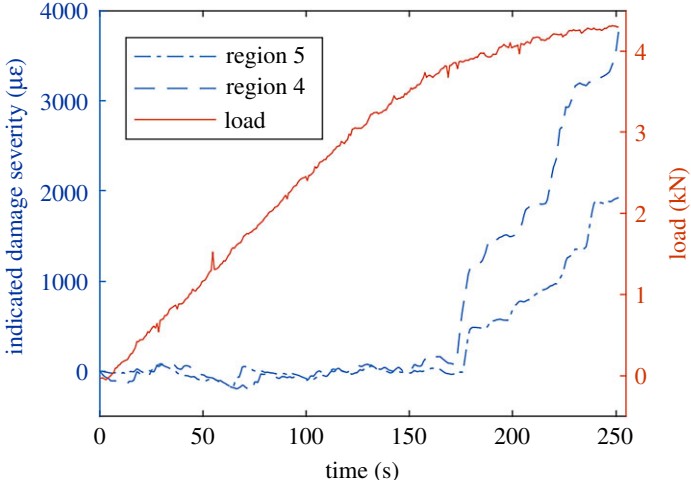

**Figure 4.** The indicated damage severity for regions no. 4 (dashed line) and no. 5 (chain line) of the CMC specimen with the specimen load signal (solid line).

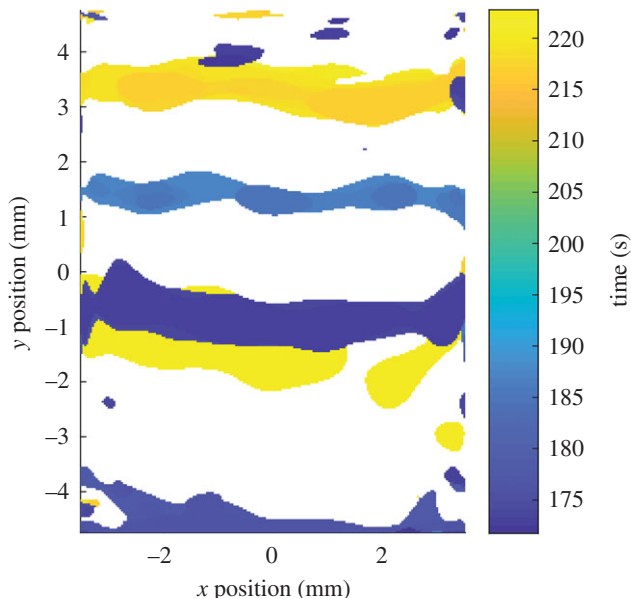

**Figure 5.** The damage-time map for region no. 4 of the CMC specimen showing the time and location at which damage was created in the specimen.

surface in the undamaged regions and at the delamination in the damaged region. In figure 10, a through-width delamination can be seen between the defective ply at the bottom of the specimen containing the waviness defect and the ply above it. A contour has been added which shows the extent of the damage indicated by the damage-time map for this specimen.

# 5. Discussion

Damage mechanics for composites is typically more complicated than those encountered in homogeneous materials such as metals or polymers. This is owing to the complexities of different materials interacting at the microscale as well as defects in the microstructure, making it difficult to predict the degradation of composites over time [23]. Techniques have been developed for the quantification of damage in components by applying a small load and then comparing the strain field in the damaged component with the strain field in a virgin component [17]. However, this cannot be used to identify the time at which damage was created nor can it identify its location. The technique used in this study enables the identification of damage occurrence by calculating the rate of change of

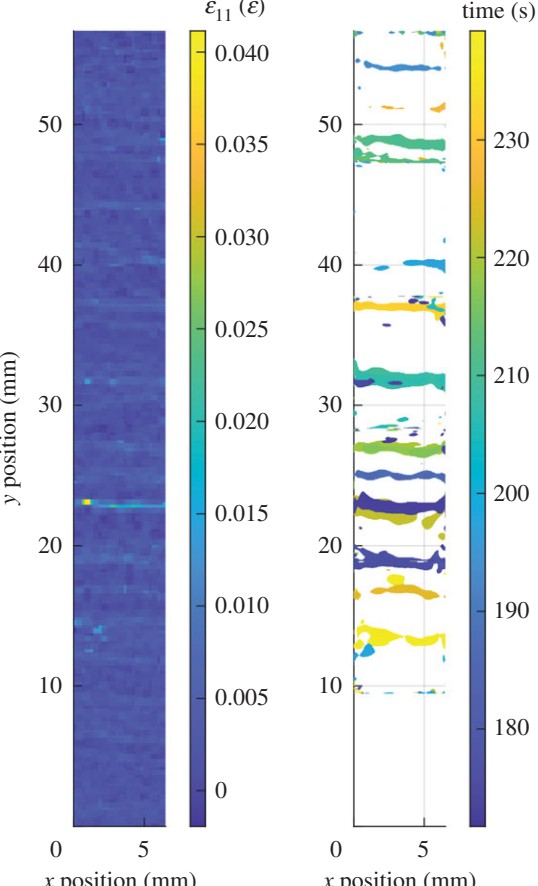

**Figure 6.** The first principal strain field on the surface of the CMC specimen just prior to failure (left) and the associated damage-time map for all regions (right).

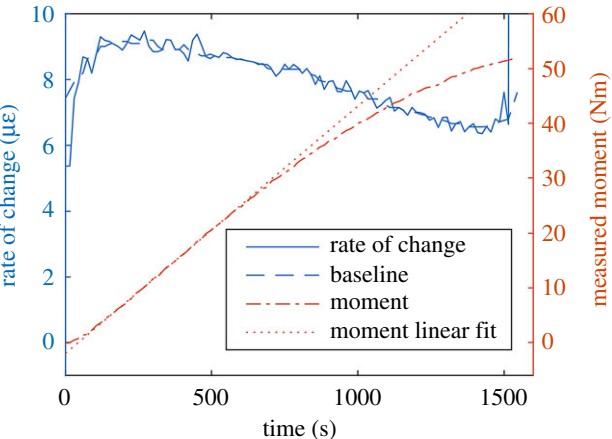

**Figure 7.** The measured bending moment for a PMC specimen containing 0% nominal waviness undergoing bending (chain line) and the rate of change of the surface strain field (solid line).

the strain field during the test and further processing it to yield an indicator of damage severity, $s_d$ obtained from equation (2.3). This allows the degradation of a component to be monitored in a way that is more effective than using the stress–strain or load–time curves. For example, the point at which damage is initiated in the CMC specimen is identifiable in the indicated damage severity plot shown in figure 4. The plot shows that the material started to exhibit damage at approximately 140 s when the load was 3.18 kN. This started in regions no. 4 and no. 5 first, before damage started to form in the other regions. As damage occurred the load was redistributed to undamaged locations which then deformed owing to the higher stresses. This resulted in the specimen becoming more compliant and

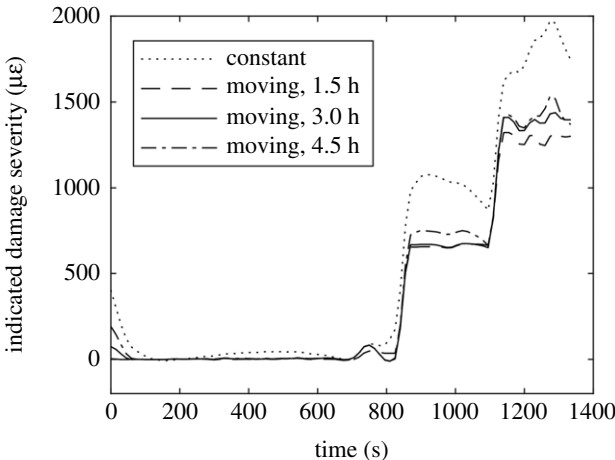

**Figure 8.** Indicated damage severity for a PMC specimen containing 25% nominal waviness. The quantity was calculated using four different approaches for quantifying the baseline rate of change: constant baseline (dotted), moving median over a period of 1.5 h (dashed), 3 h (solid) and 4.5 h (chain).

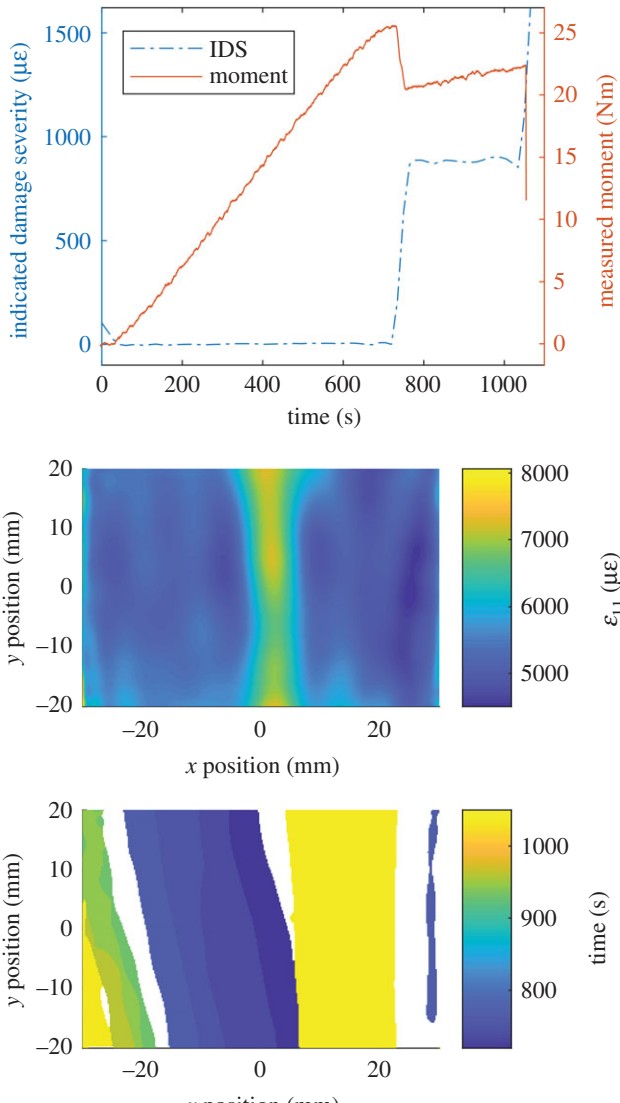

**Figure 9.** Indicated damage severity (IDS) for a PMC specimen containing a 20% nominal waviness defect (top), the first principal strain field on the specimen just prior to the proportional limit (middle) and the damage-time map for the specimen (bottom).

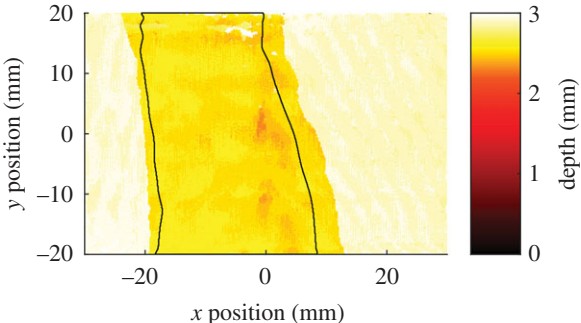

**Figure 10.** Time-of-flight ultrasound image showing a through-width delamination in a PMC specimen with a nominal waviness of 20% with the contour of the damage as measured using the strain-based damage monitoring approach superimposed (black line). The 2.9 mm thick specimen was constructed from eight plies with a nominal ply thickness of 0.36 mm.

thus the gradient of the load curve would have reduced. However, the initial damage was small and localized, and thus the changes in gradient were also small. Thus, if the creation of damage was monitored using only a load cell and extensometer, this initial damage would be undetectable owing to load cell measurement noise. Data with a similar appearance to the indicated damage severity can be obtained by monitoring the cumulative number of acoustic emission events in a component [1]. However, this requires contact with the component, making it unsuitable for tests on structures at extremes of temperature or where the mass of bonded transducers may affect the outcome of a test.

Damage events were detected by looking for sustained peaks in the indicated damage rate, $\dot{s}_d$ from equation (2.2). These events could be the initiation of matrix cracks, fibre fractures or delaminations. The algorithm presented in §2 was used to analyse these damage events to produce damage-time maps, such as those shown in figures 6 and 9. These images are a new way of presenting information about damage progression in composite components without the need to display many strain fields or view videos. This reduces the level of time and expertise required to process large amounts of experimental data. The algorithm for generating these maps is based on the concept that as damage is created it results in localized changes in the strain field. The viewing and analysis of DIC data, particularly when assessing progressive failure, often involves the experimentalist interpreting subtle changes in the shape and magnitude of patterns in strain-fields. By increasing the automation of data processing, the damage-time maps provide a less subjective method of identifying when and where damage might be occurring within a specimen. For example, the quantitative information about damage location could be used to identify, without the operator making any decisions, when damage occurs at a particular location in the specimen. The damage-time algorithm may struggle to locate delaminations that propagate slowly, as the resulting changes to the strain-field could be smaller than the measurement system uncertainty. However, the indicated damage rate would have a mean value greater than zero and thus the indicated damage severity would still increase as the delamination grows.

From the damage-time map for the CMC specimen, it was possible to identify when and where damage formed. When comparing the final strain field with this damage-time map, it is possible to see the faint bands of high strain across the width of the specimen that correspond with the damaged areas shown in the damage-time map. Acoustic emission can be used to generate images that are similar to the damage-time maps and can even identify the occurrence of damage far away from the sensors that would otherwise fall outside the region of interest for a DIC system. However, acoustic emission is typically only able to identify the location of damage events with an accuracy of 5–15 mm, with the size of sensor acting as a resolution limit [2]. By contrast, by using non-contacting DIC measurements, the spatial resolution of the damage-time maps scales with the resolution of the camera, and thus the technique described here could potentially be applied at smaller scales. At some locations, the strain field implies that damage should have occurred but it is not marked at the corresponding location in the damage-time map, for example at $y = 44$ mm in figure 6. These are likely locations where the damage event was concealed by measurement noise. The indicated damage rate data is sensitive to measurement noise and thus the damage-time maps, which are in part calculated from this data, will be negatively affected. Measurement noise has less of an effect when calculating the indicated damage severity because of the integral in equation (2.3) which helps to filter out noise in the indicated damage rate data.

The damage-time maps for the PMC specimens show different forms of damage to that encountered in the CMC specimen. The main failure mode for these specimens was delamination, which was initiated at the location of the fibre-waviness defect and spread outwards from that location. From the damage-

time maps it was possible to identify that damage spread initially in only one direction from the defect towards one of the loading noses. The loading noses were 10 mm away from each side of the region of interest and thus it can be assumed that as the delamination reached these locations it ceased to grow as the bending moment rapidly decreased to zero at the loading noses. Therefore, once the damage had reached either of the loading noses, the growth would be arrested and the delamination would start to grow from the opposite side of the defect towards the opposite loading nose. When observing the same set of specimens after failure, it was previously found [20] that the delaminations were primarily between the bottom defective ply and the ply immediately above; however, some delaminations were also observed at other ply interfaces. Because the damage-time maps are produced using only the surface strain fields, it is not possible to determine the depth at which the damage occurred and, thus, post-test inspections may still be necessary to determine the depth at which a detected damage event occurred.

The relationship between areas indicated as damaged in the damage-time map and the actual damage present in the specimen was explored for one PMC specimen using pulse-echo ultrasound to accurately measure the delamination shape. This delamination is shown in figure 10 with the damage indicated by the damage-time map overlaid as a contour. This example suggests that the damage-time algorithm is capable of identifying the location and potentially the shape of the delamination. This ability to determine damage morphology during testing is currently best achieved using computed tomography. However, computed tomography equipment is significantly more expensive than DIC equipment and restricts the speed at which tests can be conducted. For example, Bale *et al*. [5] used the Advanced Light Source, a national facility in the United States, to monitor microscale crack formation in a CMC specimen at high temperature, but each scan took 20 min to complete. Therefore, while computed tomography might yield greater information, its limitations prevent its use when testing large numbers of specimens or when the damage events occur rapidly. Damage-time maps, based on surface strain fields measured by digital image correlation, could therefore yield useful information about damage morphology without the need for expensive computed tomography equipment.

Many DIC packages now support live processing of images, such that the strain field on the surface of a component can be monitored in real-time. Currently these real-time strain fields have low frame rates, e.g. around 1 Hz; but, with increases in computer power, the frame rate is likely to approach that of the camera frame rate. The use of orthogonal decomposition to process the strain fields reduces the data dimensionality and therefore increases the computational efficiency of the strain-based monitoring algorithm. This means that each new frame can be processed to obtain the current value of the indicated damage severity as well as an updated damage-time map in just 36.4 ms (using Matlab R2017a on Windows 10 with 8 GB RAM and an i5-7500 CPU). Hence, the algorithm described in this study could be combined with real-time processing of DIC data such that the indicated damage severity data and the damage-time maps could be displayed in real-time during a test.

There are inherent drawbacks with all monitoring systems for mechanical tests, with one of the major issues being the quantities of data obtained. DIC is already an accepted technique, but the quantity of data obtained using the technique is often poorly exploited. The computationally efficient algorithms presented in this study provide new ways of processing this data, which could increase the amount of information extracted from large datasets acquired in real-time, without the need for additional effort to be expended by the operator. By using these methods, the amount of interpretation time for each experiment can be reduced, allowing engineers to perform more tests, which would lead to increased confidence in measurements, or to spend more time studying the mechanics of poorly understood processes. The outputs from these methods could also be used to explain to decision-makers, who often are non-experts, how failure occurs in components in a consistent and easy to understand manner.

## 6. Conclusion

An algorithm has been introduced for monitoring the development of damage, based on measured strain fields, and could be used to identify the time and location of damage initiation within composite components during tests. The damage that results from these events was quantified in terms of how much the strain field changed, resulting in an indicated damage severity quantity that can be calculated in real-time during tests. The developed technique was applied to data fields from digital image correlation for two distinct material systems undergoing different tests. Other measurement techniques that generate full-field stress or strain data could be used, as long as they are sensitive

enough to detect changes in surface strain owing to damage and have a high enough frame rate to monitor damage propagation. The algorithm could also be applied to measurements of surface strain on other materials that elastically deform. A novel data presentation format, using damage-time maps, was introduced that allowed information on the spatio-temporal progression of damage to be displayed. This new presentation format reduces the amount of information that must be presented to engineers in order to understand how a component failed and thus reduces the time and expertise required to interpret experimental data. This information, combined with knowledge of the microstructure of the components, could be used to guide the development of damage mechanics models leading to more accurate predictions of structural life.

Data accessibility. The experimental data can be found at the Dryad Digital Repository: https://dx.doi.org/10.5061/dryad.3s4474q [24].

Authors' contributions. W.J.R.C. performed the PMC experiments, processed the experimental data and wrote the first draft of the manuscript. J.P. performed the CMC experiments. K.A. processed the experimental data. E.A.P., K.D. and C.P. supervised the project. All authors contributed to the final manuscript.

Competing interests. We have no competing interests.

Funding. This effort was sponsored by the Air Force Office of Scientific Research, Air Force Material Command, USAF under grant no. FA9550-17-1-0272. The US Government is authorized to reproduce and distribute reprints of Governmental purpose notwithstanding any copyright notation thereon. Lt. Col. Dave Garner (EOARD) and Dr Jaimie S Tiley (AFOSR) were the programme officers for this grant. The PMC data was captured while W.J.R.C. was studying on an EPSRC Case award PhD sponsored by Airbus.

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
