## [Reviewer comments · Royal Society Open Science]

Review History

RSOS-191407.R0 (Original submission)

Review form: Reviewer 1

Is the manuscript scientifically sound in its present form?

No

Are the interpretations and conclusions justified by the results?

No

Is the language acceptable?

Yes

Do you have any ethical concerns with this paper?

No

Have you any concerns about statistical analyses in this paper?

No

Recommendation?

Major revision is needed (please make suggestions in comments)

Comments to the Author(s)

The authors present an automatic processing data for damage assessment in materials based on DIC combined with a modal analysis.

The concept is interesting and promising and I am convinced of the potential of real-time DIC post-processing to gain material insight. Having probably much less experience in experimental techniques than the authors, I focus this review on the numerical post-processing.

However the authors illustrate the agreement between ultrasound imaging and the proposed method,... but it is only an illustration, correlation is not causality and I believe that a firmer ground is needed to give confidence in the method for its actual use. The paper lacks clarity in the presentation and some details of less importance are probably given too much importance which renders the paper sometimes difficult to follow.

In its current state, the manuscript is promising but many details and maybe a firm theoretical background are missing or some analysis based on synthetic data or a modeling approach.

The use of an orthogonal decomposition (or any modal basis) is an efficient way of projecting data to a lower dimension while reducing measurement noise. Not much detail is given on the actual decomposition used in the paper. I assumed first that a singular value decomposition is applied to the different snapshots,... details are provided later but this is deterrent to the paper clarity.

It is also unclear at first whether the decomposition is applied on the displacement fields and decomposed strain fields are later deduced from these projected displacements, or if the strain fields are first computed and then decomposed in modes.

This difference has probably strong implications in terms of noise filtering.

Furthermore it is well known that orthogonal decomposition methods sometimes have issues representing localized events such as the occurrence of damage. A thorough discussion is needed concerning this point.

A thorough discussion of baseline determination is needed, two methods are briefly explained but not compared, a discussion of the median filter with or a figure comparing results with different width would be appreciated (fig 1). Other methods for estimating the baseline can be devised as it is anticipated that the first terms of the modal decomposition (which do not « see » localized events) could actually provide a very robust baseline.

To conclude I believe that the proposed method has globally a strong potential. Nevertheless the algorithmic choices and parameters of the method are not thoroughly discussed nor justified. A section dedicated to the method should be provided (separately from the experimental section). Noise filtering, mitigation and sensitivity are to be discussed (maybe not a full analysis,... but at least some insight).

For all these reasons I recommend major revisions before publication.

Minor comments:

Page 5, the discussion on h being a multiple of the time between snapshots is not necessary.

Page 6 I do not see the point of the damage severity indicator, which is has no units and is not normalized,....

Review form: Reviewer 2

Is the manuscript scientifically sound in its present form?

Yes

Are the interpretations and conclusions justified by the results?

Yes

Is the language acceptable?

Yes

Do you have any ethical concerns with this paper?

No

Have you any concerns about statistical analyses in this paper?

Yes

Recommendation?

Accept as is

Comments to the Author(s)

The authors might want to consider including the following information:

- 1) The repeatability of data analysis procedure between repeat experiments (i.e. tests conducted under identical conditions on identical sourced test coupons).
- 2) Repeatability between same process of strain measurements carried out by two or more different technicians/analysts.
- 3) correlation of the experimental findings with C(T) imaging or AE sensor output to suggest synergism between measurement methods or highlight the best technique for damage progression.

Review form: Reviewer 3

Is the manuscript scientifically sound in its present form?

Yes

Are the interpretations and conclusions justified by the results?

No

Is the language acceptable?

Yes

Do you have any ethical concerns with this paper?

No

Have you any concerns about statistical analyses in this paper?

No

Recommendation?

Major revision is needed (please make suggestions in comments)

Comments to the Author(s)

The paper contains some interesting and novel concepts regarding the application of DIC for monitoring damage progression in composite materials. However, it needs to be made clear what is new in the present work other than application to different materials over what is presented in [17] and [20] by the same group of authors.

In the paper, a methodology is described and then demonstrated on two types of composite material. The quality of the English is good, unfortunately the organisation of the paper makes it difficult to follow because the methodology is described using actual experimental results that are referred back to later in the paper.

It would be better to have one section on CMC and another on the PMC as these are completely different experiments then present the results and discussion of each together to avoid repetition. There are places in the paper where there is insufficient detail to understand what has been done and the reader is asked to refer to [17] and [20] for further information.

1. In the introduction the key benefit of using full field techniques such as DIC over point-based measurements such as strain gauges should be given. Also the global nature of a technique like AE should be mentioned as opposed to the local measurements provided by DIC which means the damage location is evident in the image data. The mention of the VFM is not clear - this requires full-field strain data from a component in a plane stress condition and can identify stiffnesses which in turn can be related to damage. At the bottom of P3 in the manuscript there is a sentence mentioning the potential for quantitative measurements of damage initiation and location being ignored, presumably in previous works, using DIC. This is a very bold statement and it is not clear exactly how the work in the paper provides something more quantitative.
2. Throughout the paper 'strain field' is mentioned but not which strain field - coordinate strains, principal strains, directions, shear strains - which is it. The plots mention epsilon₁₁ but what is 11 notating - is the y-direction strain in the plane of the sensor?
3. I could not find a mention of Figure 1 in the text prior to Figure 2 in the last paragraph on P6. Figure 1 is mentioned on P7 - the top plot shown in Figure 1 - At 210s there is also a sustained peak why is this not shaded. The damage time map mentioned in the 2nd paragraph on P7 should refer to Fig 1. Then the reader is taken to Fig 3 - where it is stated that the damaged regions had higher strain differences than the undamaged fields - this is not evident at all in the Fig 3. There are no details of loading, test set-up etc given alongside Fig 1 bottom and Fig 3 and hence it is impossible to understand their meaning. It would be much better to recast the methodology section using only Figure 2 rather than bring in the experimental results.
4. The CMC experiment is at high temperature - this is a difficult experiment for DIC - how was the speckle made - how was the specimen viewed was the camera inside the chamber??
5. For the DIC parameters please ensure the camera type - sensor type and size and lenses used are given - see page 63 in http://idics.org/wp-content/uploads/2018/10/DICGoodPracticesGuide_ElectronicVersion-V5g-181022.pdf
A tabular format with all the parameters listed such as subset size step size strain window etc. provides an easy reference for the reader rather than listing in the text
6. A diagram of the specimens is required showing the regions that the specimens were divided into and the position of the waviness etc. This will help interpret the results - for example in Figure 5 what is the difference between regions #4 and #5.
7. On P 9 an uncertainty value is provided in microstrain - how was this calculated?
8. The caption to Fig 4 mentions white dashed lines showing the 6 regions - these cannot be seen in the plot - is there a diagram missing? What is meant by accumulated damage (units microstrain) in Fig 4? Plot in Fig 4 looks very similar to plot in Fig 5 are both required?
9. In the bending tests it is not clear how the bending moment was 'measured'.
10. The discussion section is very long with the last part containing lots of speculation about what might be available in the future - this should be removed completely or reduced and moved to the conclusions section wrapping up with future work.
11. The importance of the damage time maps is not adequately described - how do these actually link to damage inception and propagation - a more through explanation is required- see remark above about more quantitative.

Decision letter (RSOS-191407.R0)

13-Nov-2019

Dear Dr Christian,

The editors assigned to your paper ("Real-time quantification of damage in structural materials during mechanical testing") have now received comments from reviewers. We would like you to revise your paper in accordance with the referee and Associate Editor suggestions which can be found below (not including confidential reports to the Editor). Please note this decision does not guarantee eventual acceptance.

Please submit a copy of your revised paper before 06-Dec-2019. Please note that the revision deadline will expire at 00.00am on this date. If we do not hear from you within this time then it will be assumed that the paper has been withdrawn. In exceptional circumstances, extensions may be possible if agreed with the Editorial Office in advance. We do not allow multiple rounds of revision so we urge you to make every effort to fully address all of the comments at this stage. If deemed necessary by the Editors, your manuscript will be sent back to one or more of the original reviewers for assessment. If the original reviewers are not available, we may invite new reviewers.

- Data accessibility

<http://datadryad.org/submit?journalID=RSOS&manu=RSOS-191407>

- **Competing interests**

- **Authors' contributions**

- **Acknowledgements**

- **Funding statement**

on behalf of Prof R. Kerry Rowe (Subject Editor)
openscience@royalsociety.org

Associate Editor's comments:

You'll note that two of the reviewers consider that you have a lot of work to do to get the paper over the line - please aim to respond effectively to each concern of the reviewers, incorporating their changes into a tracked-changes version of the manuscript on resubmission and provide a point-by-point response to their comments. Generally, authors are granted only one opportunity to revise their paper, so bear this in mind when preparing the changes. Good luck.

Comments to Author:

Reviewers' Comments to Author:

Reviewer: 1

Comments to the Author(s)

The authors present an automatic processing data for damage assessment in materials based on DIC combined with a modal analysis.

The concept is interesting and promising and I am convinced of the potential of real-time DIC post-processing to gain material insight. Having probably much less experience in experimental techniques than the authors, I focus this review on the numerical post-processing.

However the authors illustrate the agreement between ultrasound imaging and the proposed method,... but it is only an illustration, correlation is not causality and I believe that a firmer ground is needed to give confidence in the method for its actual use. The paper lacks clarity in the presentation and some details of less importance are probably given too much importance which renders the paper sometimes difficult to follow.

In its current state, the manuscript is promising but many details and maybe a firm theoretical background are missing or some analysis based on synthetic data or a modeling approach.

The use of an orthogonal decomposition (or any modal basis) is an efficient way of projecting data to a lower dimension while reducing measurement noise. Not much detail is given on the actual decomposition used in the paper. I assumed first that a singular value decomposition is applied to the different snapshots,... details are provided later but this is deterrent to the paper clarity.

It is also unclear at first whether the decomposition is applied on the displacement fields and decomposed strain fields are later deduced from these projected displacements, or if the strain fields are first computed and then decomposed in modes.

This difference has probably strong implications in terms of noise filtering.

Furthermore it is well known that orthogonal decomposition methods sometimes have issues representing localized events such as the occurrence of damage. A thorough discussion is needed concerning this point.

A thorough discussion of baseline determination is needed, two methods are briefly explained but not compared, a discussion of the median filter with or a figure comparing results with different width would be appreciated (fig 1). Other methods for estimating the baseline can be devised as it is anticipated that the first terms of the modal decomposition (which do not « see » localized events) could actually provide a very robust baseline.

To conclude I believe that the proposed method has globally a strong potential. Nevertheless the algorithmic choices and parameters of the method are not thoroughly discussed nor justified. A section dedicated to the method should be provided (separately from the experimental section). Noise filtering, mitigation and sensitivity are to be discussed (maybe not a full analysis,... but at least some insight).

For all these reasons I recommend major revisions before publication.

Minor comments:

Page 5, the discussion on h being a multiple of the time between snapshots is not necessary.

Page 6 I do not see the point of the damage severity indicator, which is has no units and is not normalized,....

Reviewer: 2

Comments to the Author(s)

The authors might want to consider including the following information:

- 1) The repeatability of data analysis procedure between repeat experiments (i.e. tests conducted under identical conditions on identical sourced test coupons).
- 2) Repeatability between same process of strain measurements carried out by two or more different technicians/analysts.
- 3) correlation of the experimental findings with C(T) imaging or AE sensor output to suggest synergism between measurement methods or highlight the best technique for damage progression.

Reviewer: 3

Comments to the Author(s)

The paper contains some interesting and novel concepts regarding the application of DIC for monitoring damage progression in composite materials. However, it needs to be made clear what is new in the present work other than application to different materials over what is presented in [17] and [20] by the same group of authors.

In the paper, a methodology is described and then demonstrated on two types of composite material. The quality of the English is good, unfortunately the organisation of the paper makes it difficult to follow because the methodology is described using actual experimental results that are referred back to later in the paper.

It would be better to have one section on CMC and another on the PMC as these are completely different experiments then present the results and discussion of each together to avoid repetition. There are places in the paper where there is insufficient detail to understand what has been done and the reader is asked to refer to [17] and [20] for further information.

1. In the introduction the key benefit of using full field techniques such as DIC over point-based measurements such as strain gauges should be given. Also the global nature of a technique like AE should be mentioned as opposed to the local measurements provided by DIC which means the damage location is evident in the image data. The mention of the VFM is not clear - this requires full-field strain data from a component in a plane stress condition and can identify stiffnesses which in turn can be related to damage. At the bottom of P3 in the manuscript there is a sentence mentioning the potential for quantitative measurements of damage initiation and location being ignored, presumably in previous works, using DIC. This is a very bold statement and it is not clear exactly how the work in the paper provides something more quantitative.
2. Throughout the paper 'strain field' is mentioned but not which strain field - coordinate strains, principal strains, directions, shear strains - which is it. The plots mention epsilon₁₁ but what is ϵ_{11} notating - is the y-direction strain in the plane of the sensor?
3. I could not find a mention of Figure 1 in the text prior to Figure 2 in the last paragraph on P6. Figure 1 is mentioned on P7 - the top plot shown in Figure 1 - At 210s there is also a sustained peak why is this not shaded. The damage time map mentioned in the 2nd paragraph on P7 should refer to Fig 1. Then the reader is taken to Fig 3 - where it is stated that the damaged regions had higher strain differences than the undamaged fields - this is not evident at all in the Fig 3. There are no details of loading, test set-up etc given alongside Fig 1 bottom and Fig 3 and hence it is impossible to understand their meaning. It would be much better to recast the methodology section using only Figure 2 rather than bring in the experimental results.
4. The CMC experiment is at high temperature - this is a difficult experiment for DIC - how was the speckle made - how was the specimen viewed was the camera inside the chamber??
5. For the DIC parameters please ensure the camera type - sensor type and size and lenses used are given - see page 63 in http://idics.org/wp-content/uploads/2018/10/DICGoodPracticesGuide_ElectronicVersion-V5g-181022.pdf

A tabular format with all the parameters listed such as subset size step size strain window etc. provides an easy reference for the reader rather than listing in the text

6. A diagram of the specimens is required showing the regions that the specimens were divided into and the position of the waviness etc. This will help interpret the results - for example in Figure 5 what is the difference between regions #4 and #5.
7. On P 9 an uncertainty value is provided in microstrain - how was this calculated?
8. The caption to Fig 4 mentions white dashed lines showing the 6 regions - these cannot be seen in the plot - is there a diagram missing? What is meant by accumulated damage (units microstrain) in Fig 4? Plot in Fig 4 looks very similar to plot in Fig 5 are both required?
9. In the bending tests it is not clear how the bending moment was 'measured'.
10. The discussion section is very long with the last part containing lots of speculation about what might be available in the future - this should be removed completely or reduced and moved to the conclusions section wrapping up with future work.
11. The importance of the damage time maps is not adequately described - how do these actually link to damage inception and propagation - a more through explanation is required- see remark above about more quantitative.

Author's Response to Decision Letter for (RSOS-191407.R0)

See Appendix A.

RSOS-191407.R1 (Revision)

Review form: Reviewer 1

Is the manuscript scientifically sound in its present form?

Yes

Are the interpretations and conclusions justified by the results?

Yes

Is the language acceptable?

Yes

Do you have any ethical concerns with this paper?

No

Have you any concerns about statistical analyses in this paper?

No

Recommendation?

Accept as is

Comments to the Author(s)

In this revised version, the authors have addressed most of the concerns and comments raised in the reviewing process of the previous version.

Minor comments:

-In page 5, when "h" is introduced it might be useful to specify that this parameter is greater than the time between frames.

-It might be interesting to discuss further the types of damage that cannot be detected by this approach which focuses localized events (in time and space). For example is it possible to follow in time the slow propagation of delamination ?

Review form: Reviewer 3

Is the manuscript scientifically sound in its present form?

Yes

Are the interpretations and conclusions justified by the results?

Yes

Is the language acceptable?

Yes

Do you have any ethical concerns with this paper?

No

Have you any concerns about statistical analyses in this paper?

No

Recommendation?

Accept as is

Comments to the Author(s)

The authors have addressed well all the comments raised by the reviewers

Decision letter (RSOS-191407.R1)

24-Jan-2020

Dear Dr Christian,

On behalf of the Editors, I am pleased to inform you that your Manuscript RSOS-191407.R1 entitled "Real-time quantification of damage in structural materials during mechanical testing" has been accepted for publication in Royal Society Open Science subject to minor revision in accordance with the referee suggestions. Please find the referees' comments at the end of this email.

The reviewers and Subject Editor have recommended publication, but also suggest some minor revisions to your manuscript. Therefore, I invite you to respond to the comments and revise your manuscript.

- Ethics statement

- Data accessibility

If you wish to submit your supporting data or code to Dryad (<http://datadryad.org/>), or modify your current submission to dryad, please use the following link:
<http://datadryad.org/submit?journalID=RSOS&manu=RSOS-191407.R1>

- Competing interests

- Authors' contributions

- Acknowledgements

- Funding statement

Because the schedule for publication is very tight, it is a condition of publication that you submit the revised version of your manuscript before 02-Feb-2020. Please note that the revision deadline will expire at 00.00am on this date. If you do not think you will be able to meet this date please let me know immediately.

To revise your manuscript, log into <https://mc.manuscriptcentral.com/rsos> and enter your Author Centre, where you will find your manuscript title listed under "Manuscripts with

Decisions". Under "Actions," click on "Create a Revision." You will be unable to make your revisions on the originally submitted version of the manuscript. Instead, revise your manuscript and upload a new version through your Author Centre.

Kind regards,
Lianne Parkhouse
Editorial Coordinator
Royal Society Open Science
openscience@royalsociety.org

on behalf of the Associate Editor, and Professor R. Kerry Rowe (Subject Editor)
openscience@royalsociety.org

Associate Editor Comments to Author:

Both reviewers are satisfied the paper is largely ready for acceptance, though the second has a couple of minor suggestions that it would be appropriate for the authors to pursue, or at least acknowledge in their manuscript.

Reviewer: 1
Comments to the Author(s)

In this revised version, the authors have addressed most of the concerns and comments raised in the reviewing process of the previous version.

Minor comments:

-In page 5, when "h" is introduced it might be useful to specify that this parameter is greater than the time between frames.

-It might be interesting to discuss further the types of damage that cannot be detected by this approach which focuses localized events (in time and space). For example is it possible to follow in time the slow propagation of delamination ?

Reviewer comments to Author:

Reviewer: 3
Comments to the Author(s)

The authors have addressed well all the comments raised by the reviewers

Author's Response to Decision Letter for (RSOS-191407.R1)

See Appendix B.

Decision letter (RSOS-191407.R2)

03-Feb-2020

Dear Dr Christian,

It is a pleasure to accept your manuscript entitled "Real-time quantification of damage in structural materials during mechanical testing" in its current form for publication in Royal Society Open Science. The comments of the reviewer(s) who reviewed your manuscript are included at the foot of this letter.

on behalf of Prof R. Kerry Rowe (Subject Editor)
openscience@royalsociety.org

Appendix A

Authors' Response to Reviewers' Comments on Manuscript RSOS-191407

We are grateful for the time the reviewers have put into reading the manuscript and for their suggestions on how it can be improved. The comments by the reviewers and our response to each request are below.

Response to Associate Editor's comments

You'll note that two of the reviewers consider that you have a lot of work to do to get the paper over the line - please aim to respond effectively to each concern of the reviewers, incorporating their changes into a tracked-changes version of the manuscript on resubmission and provide a point-by-point response to their comments. Generally, authors are granted only one opportunity to revise their paper, so bear this in mind when preparing the changes. Good luck.

- Thank you for this opportunity to improve the manuscript. We have taken great care to consider every comment made by the reviewers and have made edits to manuscript to satisfy them.

Response to Reviewer 1

The authors present an automatic processing data for damage assessment in materials based on DIC combined with a modal analysis.

The concept is interesting and promising and I am convinced of the potential of real-time DIC post-processing to gain material insight. Having probably much less experience in experimental techniques than the authors, I focus this review on the numerical post-processing.

However the authors illustrate the agreement between ultrasound imaging and the proposed method,... but it is only an illustration, correlation is not causality and I believe that a firmer ground is needed to give confidence in the method for its actual use.

- The statements regarding the correlation between shapes in the damage-time maps and associated measurements of damage have been softened. The focus is now on locating the sites of possible damage.

The paper lacks clarity in the presentation and some details of less importance are probably given too much importance which renders the paper sometimes difficult to follow.

- The structure of the paper has been modified and made more coherent, for example, by removing experimental data that was introduced before the experimental method section and increasing the focus of the discussion on the data generated by the algorithms.

In its current state, the manuscript is promising but many details and maybe a firm theoretical background are missing or some analysis based on synthetic data or a modeling approach.

- The manuscript has been updated to include a more thorough description of the key concepts behind the technique. The paper is focused on the technique for processing experimental data, a study using simulation would be a good method of exploring the correlation between damage time-maps and damage morphology; but, would move beyond the scope of this study.

The use of an orthogonal decomposition (or any modal basis) is an efficient way of projecting data to a lower dimension while reducing measurement noise. Not much detail is given on the actual decomposition used in the paper. I assumed first that a singular value decomposition is applied to the different snapshots,... details are provided later but this is deterrent to the paper clarity.

- The data was projected onto the discrete Chebyshev polynomial set. A statement has been added at the first mention of decomposition in Section 2 to make this clearer.

It is also unclear at first whether the decomposition is applied on the displacement fields and decomposed strain fields are later deduced from these projected displacements, or if the strain fields are first computed and then decomposed in modes.

- Decomposition was applied to the first (maximum) principal strain field. This has been stated in the first paragraph of Section 2. Strain was calculated from displacement data using the local regression method built into the proprietary DIC software.

This difference has probably strong implications in terms of noise filtering.

- It is possible that decomposing the displacement field first and then calculating the strain would result in reduced levels of noise. As the manuscript is focused on processing of experimental data as opposed to its capture, we chose to use the strain results generated by the DIC analysis software.

Furthermore it is well known that orthogonal decomposition methods sometimes have issues representing localized events such as the occurrence of damage. A thorough discussion is needed concerning this point.

- The local reconstruction was checked to ensure that there were no significant clusters of pixels where the difference between the reconstruction and the original strain fields were high. This has been described in Section 3.

A thorough discussion of baseline determination is needed, two methods are briefly explained but not compared, a discussion of the median filter with or a figure comparing results with different width would be appreciated (fig 1). Other methods for estimating the baseline can be devised as it is anticipated that the first terms of the modal decomposition (which do not « see » localized events) could actually provide a very robust baseline.

- A figure has been added to show how the different approaches to baseline estimation affect the indicated damage severity data. Text has been added to Section 4 to discuss this issue. There may be other methods of estimating the baseline; however, the two methods discussed in the manuscript were chosen as they relied on simple statistics.

To conclude I believe that the proposed method has globally a strong potential. Nevertheless the algorithmic choices and parameters of the method are not thoroughly discussed nor justified. A section dedicated to the method should be provided (separately from the experimental section).

- The manuscript has been modified so that the algorithm is entirely discussed in Section 2 and experimental method is only discussed in Section 3. Greater detail has been added to explain the methodology and the parameters used, with explanations that are less reliant on references.

Noise filtering, mitigation and sensitivity are to be discussed (maybe not a full analysis,... but at least some insight).

- The manuscript now contains discussion of how the measurement uncertainty of the DIC system was estimated for the PMC data and how measurement noise in the DIC data would affect the calculated quantities and maps.

For all these reasons I recommend major revisions before publication.

Minor comments:

Page 5, the discussion on h being a multiple of the time between snapshots is not necessary.

- This has been removed

Page 6 I do not see the point of the damage severity indicator, which is has no units and is not normalized,....

- This indicator has been further described in Section 2. Its units have also been clarified. Whilst strictly speaking it is dimensionless, it has the units of strain and thus is relatable to the strain fields used to calculate it. The main benefit of this indicator is now stated in Section 6.

Response to Reviewer 2

The authors might want to consider including the following information:

1) The repeatability of data analysis procedure between repeat experiments (i.e. tests conducted under identical conditions on identical sourced test coupons).

- This was the purpose of the PMC experiments, where multiple similar specimens were tested and analysed using the algorithm. This is made clearer in Section 3.2.

2) Repeatability between same process of strain measurements carried out by two or more different technicians/analysts.

- The CMC and PMC experiments were conducted by different people at different locations. This has been made clearer in section 3.2.

3) correlation of the experimental findings with C(T) imaging or AE sensor output to suggest synergism between measurement methods or highlight the best technique for damage progression.

- This would make a really interesting research project. However, as the strain-based monitoring technique is novel we have chosen to make it the focus for this manuscript and additional experiments would be needed to yield suitable CT and AE data which would require longer than the one month given for revisions to the manuscript.

Response to Reviewer 3

The paper contains some interesting and novel concepts regarding the application of DIC for monitoring damage progression in composite materials. However, it needs to be made clear what is new in the present work other than application to different materials over what is presented in [17] and [20] by the same group of authors.

- The technique described in [17] was a precursor for the indicated damage rate concept described in this manuscript, but it isn't able to characterise damage creation during a test nor is it able to identify the location of damage or give any indication of its shape. This has been stated in sections 1 and 5. Reference [20] is focused on the creation of the fibre-waviness defects. DIC was used to measure strain during testing in [20] but it was only discussed qualitatively with no algorithms used to process it. This has been stated in Section 3.

In the paper, a methodology is described and then demonstrated on two types of composite material. The quality of the English is good, unfortunately the organisation of the paper makes it difficult to follow because the methodology is described using actual experimental results that are referred back to later in the paper.

- The organisation has been changed so that experimental data is no longer referred to in section 2, which instead focuses on the developed algorithm.

It would be better to have one section on CMC and another on the PMC as these are completely different experiments then present the results and discussion of each together to avoid repetition.

- The experimental methodology section has been separated into two subsections.

There are places in the paper where there is insufficient detail to understand what has been done and the reader is asked to refer to [17] and [20] for further information.

- Greater detail has been provided on: the processing of the feature vectors, on estimation of DIC measurement uncertainty, and the experimental method used to prepare the PMC specimens.

1. In the introduction the key benefit of using full field techniques such as DIC over point-based measurements such as strain gauges should be given.

- The drawbacks of using strain gauges and the benefits of DIC based strain measurements are now discussed in Section 1.

Also the global nature of a technique like AE should be mentioned as opposed to the local measurements provided by DIC which means the damage location is evident in the image data.

- This is now discussed in Section 5.

The mention of the VFM is not clear – this requires full-field strain data from a component in a plane stress condition and can identify stiffnesses which in turn can be related to damage.

- Greater clarity has been provided at the location where the virtual fields method is mentioned.

At the bottom of P3 in the manuscript there is a sentence mentioning the potential for quantitative measurements of damage initiation and location being ignored, presumably in previous works, using DIC. This is a very bold statement and it is not clear exactly how the work in the paper provides something more quantitative.

- This statement has been significantly softened. The benefits of using this approach have been made clearer in Section 5.

2. Throughout the paper ‘strain field’ is mentioned but not which strain field – coordinate strains, principal strains, directions, shear strains – which is it. The plots mention epsilon₁₁ but what is ϵ_{11} notating – is the y-direction strain in the plane of the sensor?

- The first principal strain field was used throughout the manuscript. This is now stated in Section 2, 3, and in the captions of relevant figures.

3. I could not find a mention of Figure 1 in the text prior to Figure 2 in the last paragraph on P6.

- Due to changes in structure of the paper, Figure 1 has been modified and moved to another location. We have checked to ensure all figures are referenced in the correct order.

Figure 1 is mentioned on P7 – the top plot shown in Figure 1 – At 210s there is also a sustained peak why is this not shaded.

- The algorithm was designed to be conservative when identifying damage events, whilst the peak at 210s may be due to damage creation, it isn't sustained for the minimum time of 3 seconds, the value of h for the CMC data, required for the algorithm to automatically identify it as such. The algorithm for identifying these events is described in the penultimate paragraph of Section 2, the wording has been modified so that it doesn't imply that every peak will be identified by the algorithm as a damage event. Figure 1 (top) has been removed and the remainder of the figure moved due to changes in Section 2.

The damage time map mentioned in the 2nd paragraph on P7 should refer to Fig 1. Then the reader is taken to Fig 3 – where it is stated that the damaged regions had higher strain differences than the undamaged fields – this is not evident at all in the Fig 3. There are no details of loading, test set-up etc given alongside Fig 1 bottom and Fig 3 and hence it is impossible to understand their meaning. It would be much better to recast the methodology section using only Figure 2 rather than bring in the experimental results.

- Section 2 has been updated so that it doesn't use experimental data to provide exemplars. Examples of the concepts are now first made in section 4.

4. The CMC experiment is at high temperature – this is a difficult experiment for DIC – how was the speckle made –

- This is now described in section 3.1.

how was the specimen viewed was the camera inside the chamber??

- Neither the specimen nor the cameras were within a chamber, this is now stated in section 3.1.

5. For the DIC parameters please ensure the camera type – sensor type and size and lenses used are given – see page 63 in http://idics.org/wp-content/uploads/2018/10/DICGoodPracticesGuide_ElectronicVersion-V5g-181022.pdf

A tabular format with all the parameters listed such as subset size step size strain window etc. provides an easy reference for the reader rather than listing in the text

- As suggested, two tables of DIC hardware and software parameters have been added.

6. A diagram of the specimens is required showing the regions that the specimens were divided into and the position of the waviness etc. This will help interpret the results – for example in Figure 5 what is the difference between regions #4 and #5.

- Figure 4 was supposed to be a diagram showing the 6 regions of the CMC specimen. Unfortunately, a figure from the earliest draft of the manuscript was attached by mistake. The CMC figure has been restored and a new figure has been added to the manuscript that shows a diagram of the PMC specimens.

7. On P 9 an uncertainty value is provided in microstrain – how was this calculated?

- This was estimated via an experiment conducted using a calibration specimen. This is now discussed at the point where the uncertainty value is introduced.

8. The caption to Fig 4 mentions white dashed lines showing the 6 regions – these cannot be seen in the plot – is there a diagram missing? What is meant by accumulated damage (units microstrain) in Fig 4? Plot in Fig 4 looks very similar to plot in Fig 5 are both required?

- As mentioned previously this was a mistake by the lead author during submission. “Accumulated damage” was the name we originally gave to “indicated damage severity”. The manuscript now only uses the phrase “indicated damage severity”.

9. In the bending tests it is not clear how the bending moment was ‘measured’.

- This is now described in Section 4.

10. The discussion section is very long with the last part containing lots of speculation about what might be available in the future – this should be removed completely or reduced and moved to the conclusions section wrapping up with future work.

- The discussion is now more focused on the findings of the study as opposed to the possible directions of future work.

11. The importance of the damage time maps is not adequately described – how do these actually link to damage inception and propagation – a more through explanation is required- see remark above about more quantitative.

- Discussion of how the maps relate to damage and how they reduce the amount of subjective judgements when assessing DIC data is now discussed in Section 5.

Appendix B

Authors' Response to Reviewers' Comments on Manuscript RSOS-191407.R1

We are grateful for the time the reviewers have put into reading the revised manuscript. The reviewer comments and our response to both of them are below.

Response to Reviewer 1

In this revised version, the authors have addressed most of the concerns and comments raised in the reviewing process of the previous version.

Minor comments:

-In page 5, when "h" is introduced it might be useful to specify that this parameter is greater than the time between frames.

A sentence has been added in section 2 that clarifies this.

-It might be interesting to discuss further the types of damage that cannot be detected by this approach which focuses localized events (in time and space). For example is it possible to follow in time the slow propagation of delamination ?

This is now discussed in section 5.